# BEEP3D: Box-Supervised End-to-End Pseudo-Mask Generation for 3D Instance Segmentation

## Abstract

3D instance segmentation is crucial for understanding complex 3D environments, yet fully supervised methods require dense point-level annotations, resulting in substantial annotation costs and labor overhead. To mitigate this, box-level annotations have been explored as a weaker but more scalable form of supervision. However, box annotations inherently introduce ambiguity in overlapping regions, making accurate point-to-instance assignment challenging. Recent methods address this ambiguity by generating pseudo-masks through training a dedicated pseudo-labeler in an additional training stage. However, such two-stage pipelines often increase overall training time and complexity, hinder end-to-end optimization. To overcome these challenges, we propose BEEP3D—Box-supervised End-to-End Pseudo-mask generation for 3D instance segmentation. BEEP3D adopts a student-teacher framework, where the teacher model serves as a pseudo-labeler and is updated by the student model via an Exponential Moving Average. To better guide the teacher model to generate precise pseudo-masks, we introduce an instance center-based query refinement that enhances position query localization and leverages features near instance centers. Additionally, we design two novel losses—query consistency loss and masked-feature consistency loss—to align semantic and geometric signals between predictions and pseudo-masks. Extensive experiments on ScanNetV2 and S3DIS datasets demonstrate that BEEP3D achieves competitive or superior performance compared to state-of-the-art weakly supervised methods while remaining computationally efficient.

## 1 Introduction

3D instance segmentation (3DIS) is a fundamental task in 3D scene understanding, which involves predicting instance masks and corresponding object classes from 3D point clouds. Fully supervised 3DIS methods (Engelmann et al., 2020; Jiang et al., 2020; Chen et al., 2021; Liang et al., 2021; Schult et al., 2023) rely on dense point-wise annotations, resulting in substantial annotation costs and human effort. To alleviate this burden, recent studies have explored training 3DIS models with weaker forms of supervision, such as 3D bounding box annotations (Chibane et al., 2022; Du et al., 2023; Yu et al., 2024; Ngo et al., 2023a; Lu et al., 2024).

Among weakly supervised approaches, 3D bounding boxes are particularly attractive due to their labeling efficiency — requiring only one box per object instance. However, box-supervised 3DIS presents unique challenges compared to point-level supervision. Bounding boxes offer limited geometric detail, and overlapping regions introduce ambiguity, making it difficult to assign points to the correct instance.

To address these challenges, several methods have been proposed to generate pseudo-masks for points in ambiguous regions. Box2Mask (Chibane et al., 2022) uses rule-based heuristics to assign labels in overlapping areas. WISGP (Du et al., 2023) leverages geometric priors, such as meshes and superpoints, to propagate labels from non-overlapping to ambiguous regions. CIP-WPIS (Yu et al., 2024) employs an additional foundation 2D instance segmentation model to generate 2D masks, which are then back-projected onto 3D points. GaPro (Ngo et al., 2023a) models a posterior distribution over point labels using a Gaussian process to handle overlapping regions. BSNet (Lu et al., 2024) pre-trains

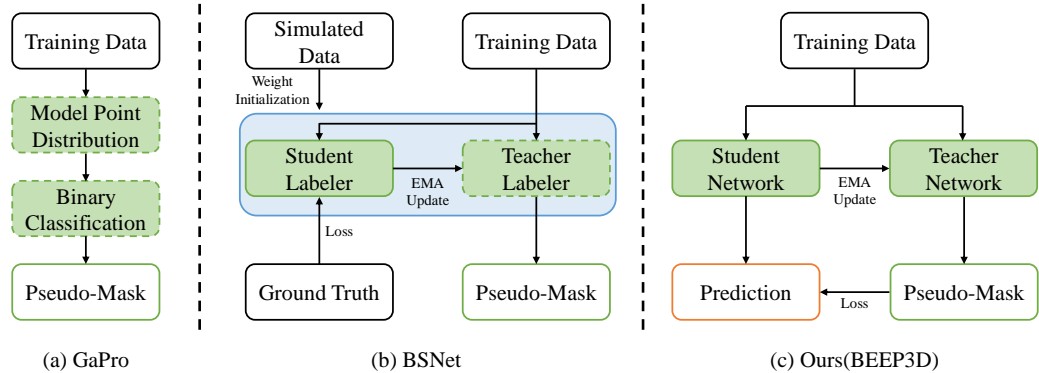

Figure 1: **Comparison of pseudo-label generation pipelines.** (a) GaPro (Ngo et al., 2023a) and (b) BSNet (Lu et al., 2024) illustrates separate pre-training stage for pseudo-label generation, whereas (c) BEEP3D (ours) generates pseudo-mask within the training loop. Dashed outlines indicate parameters frozen during training.

a pseudo-labeler on simulated data, then optimizes it using a student-teacher framework (Tarvainen & Valpola, 2017); after training, the pseudo-labeler is fixed and used to generate pseudo-masks during training the 3DIS network. However, these approaches typically require an additional training stage for the pseudo-labeler, which increases both training time and system complexity. Recent work, Sketchy-3DIS (Deng et al., 2025), jointly trains an adaptive box-to-point pseudo-labeler with a coarse-to-fine instance segmenter using intentionally perturbed sketchy bounding boxes to model annotation noise. While the method exhibits robustness to imperfect ground-truth boxes, its overall performance gains remain limited.

To overcome these limitations, we propose **BEEP3D**—Box-supervised End-to-End Pseudo-mask generation for 3D instance segmentation. BEEP3D adopts a student-teacher framework, where the teacher model acts as a pseudo-labeler and is updated via an Exponential Moving Average (EMA) of the student model, which performs the segmentation task. This structure is fully integrated into a unified training loop, enabling end-to-end optimization without requiring separate training stages. As the student model improves during training, the teacher progressively generates more accurate pseudo-masks, allowing supervision to evolve with the model. In contrast to prior methods, BEEP3D requires only minimal pre-processing of the box-supervised training data—specifically, identifying non-overlapping and overlapping regions. These design choices reduce training complexity and significantly shorten training time. Figure 1 shows that BEEP3D integrates pseudo-label generation into the training loop, rather than using a separate pre-training stage.

BEEP3D builds on a transformer-based 3DIS backbone, specifically MAFT (Lai et al., 2023), which uses two types of learnable queries: position and content queries. In our framework, however, the teacher model replaces the learnable position queries with an instance center-based query refinement that utilizes instance centers derived from box annotations. This encourages the teacher model to focus on features near object centers, leading to more accurate pseudo-mask generation. To further improve pseudo-mask generation, we introduce two consistency losses. First, the query consistency loss enforces consistency between the content queries of the student and teacher models, helping the student learn from pseudo-masks derived from instance center-refined queries. Second, the masked-feature consistency loss promotes alignment between masked point features generated by both models, ensuring consistency at the representation level and enhancing instance segmentation performance.

Our contributions are summarized as follows:

- We present **BEEP3D**, a box-supervised end-to-end framework for 3D instance segmentation that jointly optimizes a pseudo-labeler and a segmentation model within a single training loop.
- We introduce an **instance center-based query refinement** that guides the teacher model to focus on instance centers, enabling more accurate pseudo-mask generation.

- We propose a **query consistency loss** that aligns the content queries of the student and teacher models, enabling the student to learn effectively from instance-center-guided pseudo-masks.

- We introduce a **masked-feature consistency loss** that enforces alignment between masked point features from both models, promoting representation-level consistency and enhancing instance segmentation performance.

## 2 RELATED WORKS

### 2.1 3D INSTANCE SEGMENTATION

3D instance segmentation (3DIS) methods can be broadly categorized into three main paradigms: proposal-based, grouping-based, and transformer-based approaches. Proposal-based methods (Yang et al., 2019; Hou et al., 2019; Engelmann et al., 2020; Yi et al., 2019; Liu et al., 2020) follow a top-down approach, where 3D bounding boxes are first predicted and instance masks are subsequently generated within each box. The performance of these methods heavily depend on the accuracy of bounding box detection, which can significantly affect segmentation quality. Grouping-based methods (Liu & Furukawa, 2019; Jiang et al., 2020; Chen et al., 2021; Liang et al., 2021; Ngo et al., 2023b; Vu et al., 2022; Wu et al., 2022; Zhong et al., 2022) adopt a bottom-up strategy, predicting semantic categories and geometric offsets for each point, followed by clustering to form instances. However, their performance is often constrained by the reliability of the clustering process.

To address these limitations, transformer-based methods have recently gained popularity. These models directly predict instance masks by computing similarities between learnable queries and point features, eliminating the need for explicit proposal generation or clustering. Transformer-based methods (Schult et al., 2023; Sun et al., 2023; Lu et al., 2023; Lai et al., 2023; Kolodiazhnyi et al., 2024) represent each instance via learnable queries and generate instance masks using a transformer decoder, allowing the model to capture global scene context. While fully supervised transformer-based methods have demonstrated strong performance, they rely on dense point-level annotations, which are expensive and time-consuming to acquire. In contrast, our proposed method, BEEP3D, adopts a weakly supervised approach that eliminates the need for point-level labels, significantly reducing annotation costs.

### 2.2 WEAKLY SUPERVISED 3D INSTANCE SEGMENTATION

Weakly supervised 3DIS methods can be categorized by their supervision types, including sparse-point annotations and 3D bounding box annotations. Sparse-point approaches (Hou et al., 2021; Xie et al., 2020) leverage a small subset of labeled points to supervise model training. In contrast, 3D bounding box annotations (Chibane et al., 2022; Du et al., 2023; Yu et al., 2024; Ngo et al., 2023a; Lu et al., 2024; Deng et al., 2025) provide coarse yet structured supervision, including approximate object locations and sizes, making them a practical alternative to point-level labels. However, overlapping bounding boxes can lead to significant ambiguity in point-to-instance assignment.

Box2Mask (Chibane et al., 2022) uses rule-based heuristics to assign labels in overlapping regions and predicts instance masks by estimating box parameters for each point, followed by Hough voting and non-maximum clustering. WISGP (Du et al., 2023) leverages geometric priors, such as meshes and superpoints, to generate initial pseudo-masks for training. The network is first trained using these geometry-derived labels, and then refined using its own predictions as supervision in a self-training fashion. CIP-WPIS (Yu et al., 2024) employs the Segment Anything Model (SAM) (Kirillov et al., 2023) to predict 2D masks from multiple views and projects them onto 3D points to obtain pseudo-masks. GaPro (Ngo et al., 2023a) models the posterior distribution of ambiguous points using a Gaussian process, which acts as a binary classifier to assign labels in overlapping regions. BSNet (Lu et al., 2024) pre-trains a pseudo-labeler (SAFormer) on simulated data, and then jointly optimizes it with a student model using a Mean Teacher framework (Tarvainen & Valpola, 2017). After optimization, the pseudo-labeler is frozen and used to generate pseudo-masks for 3DIS training. Sketchy-3DIS (Deng et al., 2025) couples an adaptive box-to-point pseudo-labeler with a coarse-to-fine instance segmenter and trains with "sketchy" (perturbed) bounding boxes, showing robustness to box-level noise.

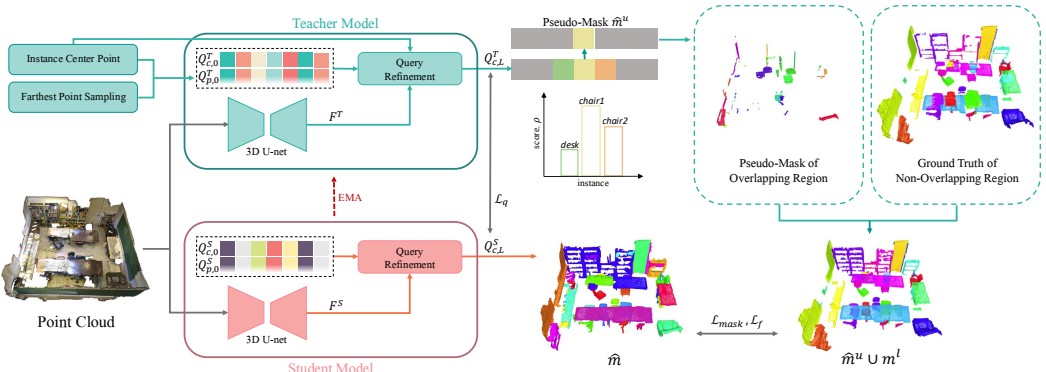

Figure 2: **Overview of BEEP3D.** The student and teacher models share the same architecture, consisting of a 3D U-Net backbone and transformer decoder layers with position and content queries. The teacher model generates pseudo-masks for overlapping regions using instance centers and FPS, while the student is supervised by the union of pseudo-masks $\widehat{m}^u$ and ground-truth masks $m^l$ from box annotations.

Most of these approaches introduce an additional training stage for the pseudo-labeler and rely on complex prior information, such as geometric structures (e.g., meshes, superpoints), probabilistic models, or simulated data. However, the need for separate training and heavy pre-processing increases system complexity and hinders end-to-end optimization. BEEP3D addresses these limitations by introducing an end-to-end trainable framework that generates high-quality pseudo-masks from box annotations with minimal pre-processing.

## 3 METHOD

Following prior work (Chibane et al., 2022; Ngo et al., 2023a), we categorize the point labels derived from box annotations into two types: (1) $m^u \in \mathbb{R}^{N_u \times K}$, representing points located in overlapping regions of multiple boxes, for which the instance labels are considered undetermined; (2) $m^l \in \mathbb{R}^{N_l \times K}$, representing points located within a single box, which can be directly assigned ground-truth labels based on the corresponding instance box. Let $P^u$ and $P^l$ denote the sets of points in the overlapping and non-overlapping regions, respectively. We define $N_u = |P^u|$ and $N_l = |P^l|$ as the number of points in each region type, and $K$ as the number of ground-truth instances in the scene.

Figure 2 illustrates an overview of our proposed framework, BEEP3D, which adopts a student-teacher framework (Tarvainen & Valpola, 2017). Both the student and teacher models share the same network structure, based on MAFT (Lai et al., 2023). The teacher model serves as a pseudo-labeler, generating pseudo-masks $\widehat{m}^u$ for points in the overlapping regions, and is updated via an Exponential Moving Average (EMA) of the student model parameters. The student model learns instance segmentation by supervising on the union of the pseudo-masks $\widehat{m}^u$ and the ground-truth masks $m^l$ from the non-overlapping regions.

### 3.1 PROBLEM SETUP

The input point cloud is denoted as $P = \{p_j\}_{j=1}^{N} \in \mathbb{R}^{N \times 6}$, where each point $p_j = (x_j, y_j, z_j, r_j, g_j, b_j) \in \mathbb{R}^6$ contains 3D coordinates and RGB color values. Although MAFT performs voxelization and superpoint-level pooling during feature extraction, the resulting features are eventually mapped back to individual points. Therefore, for notational simplicity, we represent all features in a point-wise manner. We denote the point-wise features from the student and teacher networks as $F^S, F^T \in \mathbb{R}^{N \times C}$, where $C$ is the feature dimension. MAFT adopts two types of queries for refinement: position queries and content queries. The position queries represent spatial priors of instances, while the content queries encode instance-aware information. Let $Q_{c,t}^S, Q_{c,t}^T \in \mathbb{R}^{N_Q \times C}$ and $Q_{p,t}^S, Q_{p,t}^T \in \mathbb{R}^{N_Q \times 3}$ denote the content and position queries of the student and teacher models

at refinement step $t \in \{0, 1, \ldots, L\}$, where $N_Q$ is the number of queries and $L$ is the number of transformer decoder layers.

These queries are refined through a sequence of $L$ transformer decoder layers using both self-attention and cross-attention mechanisms. For the student model, the query decoder outputs instance masks $\widehat{m}$ and corresponding semantic classes based on the refined content queries $Q_{c,L}^S$, following the standard MAFT pipeline. The process of generating pseudo-masks from the teacher model is described in the next subsection.

### 3.2 Pseudo-mask Generation

Following prior works on transformer-based 3D instance segmentation (Schult et al., 2023; Sun et al., 2023; Lai et al., 2023), we compute similarity scores between the refined content queries and the point-wise features as:

$$\rho = \sigma(Q_{c,L}^T \cdot F^{T\top}) \in \mathbb{R}^{N_Q \times N}, \tag{1}$$

where $\sigma(\cdot)$ denotes the sigmoid function, and each element $\rho_{ij}$ represents the similarity between the $i$-th query and the feature of point $p_j$. Since the number of queries $N_Q$ does not necessarily match the number of object instances $K$ in a scene, we apply the Hungarian Algorithm (Kuhn, 1955) to find an optimal bipartite matching between queries and ground-truth boxes. The resulting similarity matrix after matching is denoted as $\rho' \in \mathbb{R}^{K \times N}$, where each row corresponds to a matched query.

We formulate the pseudo-mask generation for points in overlapping regions $P^u \subset P$ for $i$-th instance as follows. Each point $p_j \in P^u$ is assigned to the most similar matched query among those whose ground-truth instance contains the point:

$$a_j := \arg\max_{l \in \{i \mid m_{ij}^u = 1\}} \rho'_{lj}, \quad \text{for} \quad \forall p_j \in P^u \tag{2}$$

$$P_i^u := \{p_j \in P^u \mid a_j = i\}, \tag{3}$$

$$\widehat{m}_i^u = [\mathbb{I}[p_j \in P_i^u]]_{j=1}^{N^u} \in \{0,1\}^{N^u}, \tag{4}$$

where $a_j$ denotes the instance index assigned to point $p_j$ among valid overlapping candidates, $P_i^u$ is the set of overlapping points assigned to instance $i$, and $\widehat{m}_i^u$ is the corresponding binary pseudo-mask. As the teacher model assigns pseudo-masks only to points within overlapping regions, we limit the candidate instances to those whose boxes include the given point $p_j$. Similarity scores from other unmatched queries or irrelevant instances are ignored during mask construction.

### 3.3 Instance Center–based Query Refinement

Unlike prior works that employ learnable parameters for query refinement, we refine the position queries in the teacher model using instance center coordinates derived from box annotations.

We first sample $N_Q$ normalized point coordinates using Farthest Point Sampling (FPS) (Qi et al., 2017), denoted as $P_{\text{FPS}} \in \mathbb{R}^{N_Q \times 3}$. The set of instance centers is defined as $C \in \mathbb{R}^{K \times 3}$. Since the number of boxes varies across scenes, we compute attention weights between the sampled points $P_{\text{FPS}}$ and instance centers $C$, and aggregate the centers using a weighted sum:

$$Q_{p,0}^T = \text{softmax}(P_{\text{FPS}} \cdot C^\top) \cdot C. \tag{5}$$

After the first query refinement step, the teacher model updates the position queries at each timestep $t > 0$ as:

$$Q_{p,t}^T \leftarrow \text{softmax}(Q_{p,t}^T \cdot C^\top) \cdot C. \tag{6}$$

This strategy allows the model to aggregate a variable number of instance centers into each position query, guided by FPS positions that cover the scene globally. Through self-attention and cross-attention in the subsequent transformer layers, the teacher model can effectively focus on features around instance centers, leading to the generation of more accurate pseudo-masks.

### 3.4 Consistency Loss for Student-Teacher Framework

We introduce two consistency losses that encourage the student model to align its representations with those of the teacher model, thereby improving the effectiveness of pseudo-mask supervision.

**Query Consistency Loss**    To facilitate effective guidance from the teacher model, we design a query consistency loss, $\mathcal{L}_q$, between the refined content queries of the teacher and student models. The distance between the student content query $Q_{c,L}^S$ and the teacher content query $Q_{c,L}^T$ is minimized with an $L1$ loss:

$$\mathcal{L}_q = \|Q_{c,L}^S - Q_{c,L}^T\|_1. \tag{7}$$

This loss encourages the student model to absorb instance-aware cues refined by the teacher model based on instance centers, as described in Section 3.3.

**Masked-Feature Consistency Loss**    To enforce alignment at the representation level, we propose a masked-feature consistency loss $\mathcal{L}_f$. We compute the mean of masked point features for both teacher and student models.

For the teacher model, the masked-feature for the $k$-th instance is computed as:

$$F_k'^T = \frac{1}{n_k^T} \sum_{j=1}^{N} (m_k^l \cup \widehat{m}_k^u)_j \cdot f_j^T, \quad \text{where } n_k^T = \sum_{j=1}^{N} (m_k^l \cup \widehat{m}_k^u)_j, \tag{8}$$

where $f_j^T$ denotes the feature of point $p_j$ from the teacher model, and $k \in \{1, \dots, K\}$ indexes ground-truth instances.

Similarly, for the student model, we use the predicted instance masks:

$$F_k'^S = \frac{1}{n_k^S} \sum_{j=1}^{N} (\widehat{m}_k)_j \cdot f_j^S, \quad \text{where } n_k^S = \sum_{j=1}^{N} (\widehat{m}_k)_j, \tag{9}$$

where $f_j^S$ is the feature of point $p_j$ from the student model.

The masked-feature consistency loss is then computed as the $L_2$ distance between the aggregated features over all instances:

$$\mathcal{L}_f = \sum_{k=1}^{K} \|F_k'^T - F_k'^S\|_2^2. \tag{10}$$

This loss promotes feature-level consistency between the student and teacher models, helping the student learn more accurate instance representations.

### 3.5    TRAINING A 3DIS NETWORK WITH PSEUDO-MASK

The student model is trained to predict instance segmentation masks $\widehat{m}$ by minimizing the loss against the target masks $m = \widehat{m}^u \cup m^l$, where $\widehat{m}^u$ is the pseudo-mask for overlapping regions and $m^l$ is the ground-truth mask for non-overlapping regions.

The instance masks are supervised using a combination of binary cross-entropy loss $\mathcal{L}_{bce}$, dice loss $\mathcal{L}_{dice}$, and classification loss $\mathcal{L}_{cls}$ based on cross-entropy. The overall mask loss $\mathcal{L}_{mask}$ is defined as:

$$\mathcal{L}_{mask} = \lambda_{bce}\mathcal{L}_{bce} + \lambda_{dice}\mathcal{L}_{dice} + \lambda_{cls}\mathcal{L}_{cls}, \tag{11}$$

The final training objective combines the mask loss with the proposed consistency losses:

$$\mathcal{L} = \mathcal{L}_{mask}(\widehat{m}, m) + \lambda_q\mathcal{L}_q + \lambda_f\mathcal{L}_f. \tag{12}$$

## 4    EXPERIMENTS

### 4.1    DATASETS AND METRICS

We evaluate BEEP3D on two standard benchmarks: ScanNetV2 (Dai et al., 2017) and S3DIS (Armeni et al., 2016). ScanNetV2 contains 1,201 scans for training, 312 for validation, and 100 for testing, covering 18 object classes. S3DIS consists of 271 rooms across six areas, annotated with 13 semantic categories. Following the standard protocol, Area 5 is used for validation, and the remaining areas are used for training. We report average precision (AP), as well as $AP_{50}$ and $AP_{25}$ on ScanNetV2. On S3DIS, we report AP and $AP_{50}$. The AP metric, commonly used in 3D instance segmentation, averages precision over IoU thresholds from 50% to 95% in 5% increments. The $AP_{50}$ and $AP_{25}$ scores correspond to fixed IoU thresholds of 50% and 25%, respectively. For ScanNetV2, we report test set performance via submission to the official benchmark server.

Table 1: Comparison of box-supervised 3D instance segmentation performance with previous works on the ScanNetV2 validation set and test set. **% Full** indicates the percentage of the AP score relative to the corresponding fully supervised method. Fully supervised methods are marked as  Fully .

| Method | Validation set | | | | Test set | | | |
|---|---|---|---|---|---|---|---|---|
| | AP | % full | $AP_{50}$ | $AP_{25}$ | AP | % full | $AP_{50}$ | $AP_{25}$ |
| ISBNet (Ngo et al., 2023b) | 54.5 | - | 73.1 | 82.5 | 55.9 | - | 76.3 | 84.5 |
| Mask3D (Schult et al., 2023) | 55.2 | - | 73.7 | 83.5 | 56.6 | - | 78.0 | 87.0 |
| SPFormer (Sun et al., 2023) | 56.3 | - | 73.9 | 82.9 | 54.9 | - | 77.0 | 85.1 |
| MAFT (Lai et al., 2023) | 58.4 | - | 75.9 | 84.5 | 57.8 | - | 77.4 | - |
| OneFormer3D (Kolodiazhnyi et al., 2024) | 59.3 | - | 78.1 | 86.4 | 56.6 | - | 80.1 | 89.6 |
| Box2Mask (Chibane et al., 2022) | 39.1 | - | 59.7 | 71.8 | 43.3 | - | 67.7 | 80.3 |
| GaPro (Ngo et al., 2023a) + ISBNet (Ngo et al., 2023b) | 50.6 | 92.8% | 69.1 | 79.3 | 49.3 | 88.2% | 69.8 | 81.0 |
| GaPro (Ngo et al., 2023a) + SPFormer (Sun et al., 2023) | 51.1 | 90.8% | 70.4 | 79.9 | 48.2 | 87.7% | 69.2 | 82.4 |
| BSNet (Lu et al., 2024) + ISBNet (Ngo et al., 2023b) | 52.8 | 96.9% | 71.6 | 82.6 | - | - | - | - |
| BSNet (Lu et al., 2024) + SPFormer (Sun et al., 2023) | 53.3 | 90.8% | 72.7 | 83.4 | - | - | - | - |
| BSNet (Lu et al., 2024) + MAFT (Lai et al., 2023) | 56.2 | 96.2% | **75.9** | **85.7** | - | - | - | - |
| Sketchy-3DIS (Deng et al., 2025) | - | - | 68.8 | 83.6 | - | - | 70.1 | **86.6** |
| Ours + SPFormer (Sun et al., 2023) | 55.2 | 98.0% | 73.6 | 83.3 | - | - | - | - |
| Ours + MAFT (Lai et al., 2023) | **57.3** | **98.1**% | 75.3 | 84.3 | **53.0** | **91.7**% | **73.2** | 84.6 |
| Ours + OneFormer3D (Kolodiazhnyi et al., 2024) | 55.8 | 94.1% | 74.7 | 84.6 | - | - | - | - |

Table 2: Comparison of 3D instance segmentation performance with previous works on the S3DIS dataset (Area 5). **Sup.** denotes the type of supervision used by each method.

| Sup. | Method | AP | $AP_{50}$ |
|---|---|---|---|
| Mask | SoftGroup(Vu et al., 2022) | 51.6 | 66.1 |
| | ISBNet (Ngo et al., 2023b) | 54.0 | 65.8 |
| | Mask3D (Schult et al., 2023) | 56.6 | 68.4 |
| | SPFormer (Sun et al., 2023) | - | 66.8 |
| | MAFT (Lai et al., 2023) | - | 69.1 |
| Box | Box2Mask (Chibane et al., 2022) | 43.6 | 54.6 |
| | GaPro(Ngo et al., 2023a) + SoftGroup(Vu et al., 2022) | 47.0 | 62.1 |
| | GaPro(Ngo et al., 2023a) + ISBNet(Ngo et al., 2023b) | 50.5 | 61.2 |
| | BSNet(Lu et al., 2024) + SoftGroup(Vu et al., 2022) | 51.4 | 62.8 |
| | BSNet(Lu et al., 2024) + ISBNet(Ngo et al., 2023b) | 53.0 | 64.3 |
| | Sketchy-3DIS(Deng et al., 2025) | 53.4 | **69.1** |
| | Ours + ISBNet(Ngo et al., 2023b) | 51.3 | 64.4 |
| | Ours + MAFT(Lai et al., 2023) | **53.6** | 67.3 |

### 4.2 IMPLEMENTATION DETAILS

For both ScanNetV2 and S3DIS, we adopt a 5-layer U-Net as the feature extraction backbone. We apply a voxelization size of 2 cm and use 6 transformer decoder layers. For the weights of the losses, $(\lambda_{bce}, \lambda_{dice}, \lambda_{cls}, \lambda_q, \lambda_f)$ are set to $(1.0, 1.0, 0.5, 0.5, 0.5)$ for ScanNetV2, and $(5.0, 1.0, 2.0, 2.0, 2.0)$ for S3DIS. We use an EMA decay rate of 0.99 and describe the scheduling strategy in the appendix. We train all models for 512 epochs using the AdamW optimizer (Loshchilov & Hutter, 2017) and the one-cycle learning rate scheduler (Smith & Topin, 2019), with an initial learning rate of 1e-4. All the models are trained from scratch on each dataset. All experiments are conducted on a single NVIDIA RTX 3090 GPU, unless stated otherwise. We additionally implement our framework on SPFormer (Sun et al., 2023), OneFormer3D (Kolodiazhnyi et al., 2024) for ScanNetV2 and ISBNet (Ngo et al., 2023b) for S3DIS. Corresponding implementation details are provided in the appendix.

### 4.3 COMPARISON TO PREVIOUS WORK

**ScanNetV2**   Table 1 compares BEEP3D with prior methods on the ScanNetV2 validation and test sets. BEEP3D achieves the highest AP on both sets among box-supervised approaches, supported by an instance-center-based query-refinement module and two consistency losses. Despite using only box-level annotations, BEEP3D reaches 98.1% (MAFT), 98.0% (SPFormer), and 94.1% (OneFormer3D)

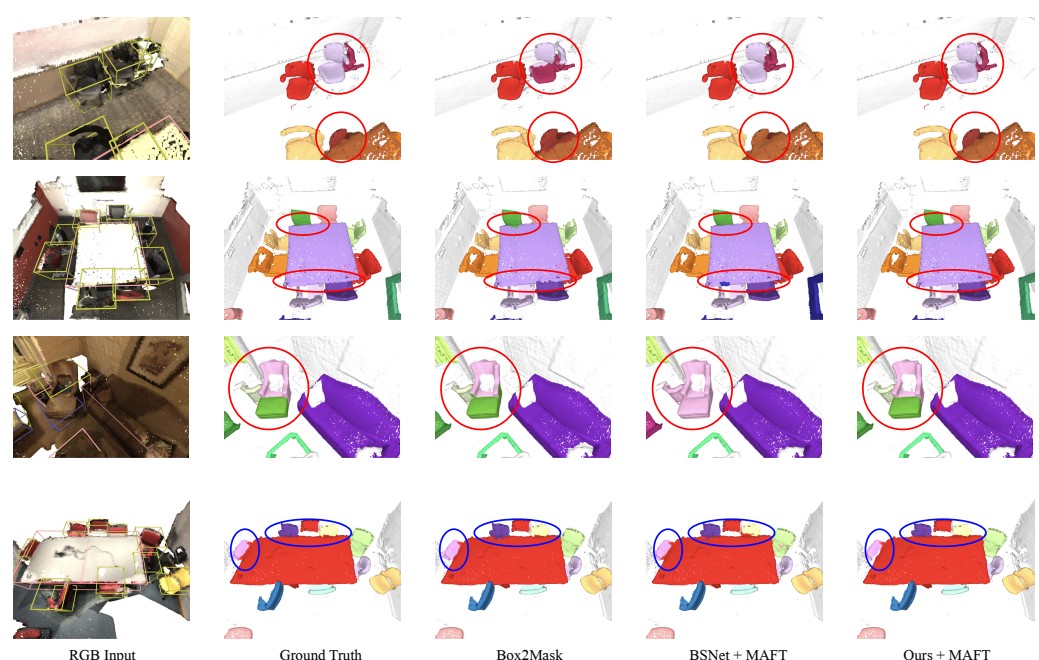

RGB Input     Ground Truth     Box2Mask     BSNet + MAFT     Ours + MAFT

Figure 3: **Visualization results on ScanNetV2 validation set.** The first column shows the RGB input with ground-truth 3D bounding boxes (in yellow), and the second column presents the annotated instance masks. Highlighted regions (in circles) indicate overlapping areas, where box-supervised methods frequently struggle due to ambiguity in point-to-instance assignment.

of the AP achieved by their corresponding fully supervised counterparts on the validation set. These results further demonstrate the generality of BEEP3D across the transformer-based 3DIS architectures. While BEEP3D leads at high IoU thresholds, it remains competitive at lower thresholds—ranking second at $AP_{50}$ and $AP_{25}$—which highlights its overall robustness. We further evaluate statistical significance by training with five different random seeds, obtaining consistent performance ($57.2 \pm 0.4AP$). Detailed results are included in the appendix. On the test set, BEEP3D attains state-of-the-art AP and $AP_{50}$ among box-supervised methods, with a clear margin over prior work.

**S3DIS** Table 2 reports results on the S3DIS Area 5 split. When integrated with MAFT, BEEP3D achieves the highest AP, outperforming prior state-of-the-art box-supervised methods and demonstrating both effectiveness and cross-dataset generalization. For a fair comparison with prior box-supervised works that use the same backbone (Ngo et al., 2023a; Lu et al., 2024), we additionally evaluate BEEP3D with ISBNet (Ngo et al., 2023b) following its corresponding training protocol. As our key components are tailored to transformer-based architectures, the ISBNet variant yields negligible improvements.

**Qualitative Comparisons** Figure 3 shows qualitative results on the ScanNetV2 validation set, comparing BEEP3D, BSNet, Box2Mask, and the ground-truth annotations. BEEP3D and BSNet are both implemented on the MAFT backbone. Our method produces visually accurate masks, particularly around object boundaries and in regions with overlapping instances, where box-supervised methods often struggle.

## 4.4 ABLATION STUDY

We perform ablation studies on ScanNetV2 validation set to assess the contribution of each component in BEEP3D.

**Effect of Consistency Losses** Table 3 presents a comparative analysis of the impact of the proposed consistency losses on segmentation performance. The results show that adding both the query

Table 3: Ablation study on the effect of each loss component in our framework.

| $\mathcal{L}_q$ | $\mathcal{L}_f$ | AP | $AP_{50}$ | $AP_{25}$ |
|---|---|---|---|---|
| | | 54.6 | 72.6 | 82.7 |
| ✓ | | 55.1 | 73.5 | 82.8 |
| | ✓ | 54.9 | 73.5 | 82.9 |
| ✓ | ✓ | **57.3** | **75.3** | **84.3** |

Table 4: Effect of instance center–based query refinement. **mACC** denotes the mean accuracy of pseudo-masks in the overlapping regions. [†] indicates results reproduced from BSNet (Lu et al., 2024).

| Method | mACC | AP | $AP_{50}$ | $AP_{25}$ |
|---|---|---|---|---|
| GaPro[†] | 38.1 | - | - | - |
| BSNet[†] | 59.6 | - | - | - |
| Ours + MAFT (Lai et al., 2023) | | | | |
| w/ instance center | **79.9** | **57.3** | **75.3** | **84.3** |
| w/o instance center | 73.4 | 53.9 | 72.3 | 82.2 |

Table 5: Ablation study on the effect of EMA update and direct self-training loop.

| Method | AP | $AP_{50}$ | $AP_{25}$ |
|---|---|---|---|
| Self-training | 53.4 | 72.9 | 82.8 |
| EMA Update (Ours) | **57.3** | **75.3** | **84.3** |

consistency loss and the masked-feature consistency loss significantly improves overall performance, demonstrating the necessity and effectiveness of each component.

**Effect of Instance Center-based Query Refinement** Table 4 evaluates the effectiveness of our instance center-based query refinement. We first assess the quality of pseudo-masks on the ScanNetV2 training set using mean accuracy (mACC) (Lu et al., 2024), which measures the average accuracy of binary pseudo-masks in overlapping regions. To contextualize the improvement, we also compare with prior methods. Our method with instance center-based query refinement achieves the highest mACC, significantly outperforming existing approaches. The pseudo-labeler captures features near instance centers, producing more precise pseudo-masks and higher AP than without query refinement.

**Effect of EMA Update Framework** To validate the necessity of the EMA-based teacher–student framework, we compare our default EMA update with a direct self-training baseline, where the student at step $N$ generates pseudo-masks to supervise training at step $N + 1$. As shown in Table 5, removing the EMA teacher leads to a clear drop in performance. This demonstrates that a purely self-training loop cannot provide reliable supervision under box-only annotations: without EMA, pseudo-labels remain noisy and unstable, particularly in overlapping regions. In contrast, the EMA teacher acts as a temporally smoothed model that co-evolves with the student, generating progressively refined pseudo-masks throughout training. This stabilization effect enables the student to learn from higher-quality pseudo-labels than those produced by any individual model snapshot, explaining the consistent gains observed with the EMA update.

## 4.5 TRAINING TIME AND PARAMETER EFFICIENCY

Table 6 compares the time for 3DIS training (T) and the additional time for pseudo-label generation (T′). All times are measured on the ScanNetV2 training set using a single NVIDIA RTX 3090 GPU. When the base model(SPFormer) is identical, the training time (T) is generally similar across methods. However, GaPro requires more time due to an additional self-training stage following the initial training. Comparing T′, GaPro includes both the initial pseudo-label generation and the regeneration used for self-training. For BSNet, T′ includes simulated-sample generation, SAFormer optimization, and pseudo-label generation. In contrast, BEEP3D optimizes the pseudo-labeler and

Table 6: Comparison of 3DIS training time (T) and psuedo-label generation time (T′) on ScanNetV2 training set. [†] indicates the result reported from BSNet (Lu et al., 2024).

| Method | T (hrs) | T′ |
|---|---|---|
| GaPro (Ngo et al., 2023a) + SPFormer (Sun et al., 2023) | 47.8 | 13.0 |
| BSNet (Lu et al., 2024) + SPFormer (Sun et al., 2023) | 27.1 | 7.5[†] |
| Ours + SPFormer (Sun et al., 2023) | 29.4 | - |
| Ours + MAFT (Lai et al., 2023) | 35.6 | - |

dynamically generates pseudo-labels without a separate pre-training stage, so no additional T′ is required, demonstrating ease of training and time efficiency.

## 5 CONCLUSIONS

In this paper, we presents **BEEP3D**, the end-to-end framework for pseudo-mask generation in box-supervised 3D instance segmentation. BEEP3D adopts a student-teacher architecture, where the teacher model serves as a pseudo-labeler and the student model learns to predict instance segmentation masks. Our instance center-based query refinement strategy enables the teacher model to generate more accurate pseudo-masks, while the proposed consistency losses effectively guide the student model to align with the teacher model. BEEP3D achieves improved training efficiency over prior methods while maintaining competitive segmentation performance, nearly matches its fully supervised counterparts.

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

# A  APPENDIX

## A1  MORE IMPLEMENTATION DETAILS

For transformer-based 3DIS backbone (MAFT (Lai et al., 2023), SPFormer (Sun et al., 2023), OneFormer3D (Kolodiazhnyi et al., 2024)), we use 6 transformer decoder layers with 8 attention heads. The hidden dimension is set to 256, and the feed-forward dimension is 1024. The content queries for both the student and teacher models are initialized to zero, while the position queries of the student model are initialized with learnable random parameters. These initialization strategies have been shown to be effective in MAFT. Following the relative position encoding scheme introduced in MAFT, the position queries are also used for the positional encoding of the content queries.

**SPFormer Implementation**  Unlike MAFT, which refines both position and content queries, SP-Former handles only content queries. To incorporate our instance center–based query refinement into SPFormer, we use the initialized position query $Q_{p,0}^T$, defined in Equation (5), as the positional encoding for the teacher model. Following prior works (Schult et al., 2023; Lai et al., 2023), we encode $Q_{p,0}^T$ using Fourier positional encodings. For the student model, the positional encoding is implemented using learnable random parameters, consistent with our MAFT-based implementation.

**OneFormer3D Implementation**  OneFormer3D (Kolodiazhnyi et al., 2024) adopts the same transformer-based architecture as SPFormer, but uses a set of queries consisting of both instance and semantic queries. Therefore, we apply our instance center–based query refinement to the full query set in OneFormer3D in the same way as in SPFormer.

**ISBNet Implementation**  Unlike other transformer-based 3DIS networks, ISBNet (Ngo et al., 2023b) samples $K$ instance candidates and updates them via point-aggregation layers to obtain instance kernels. To incorporate our instance center–based query refinement into this pipeline, we apply the refinement directly within the point-aggregation layers: each candidate feature is treated as a content query, and its corresponding coordinates serve as the positional query for the teacher model. For the student model, we follow the original ISBNet implementation without modification.

**EMA Scheduling**  We also provide additional details on the EMA decay-rate scheduling implementation. We start with a small EMA decay and progressively increase it toward the target value $\alpha$ as training proceeds:

$$\alpha_t = \min\left(1 - \frac{1}{t+1}, \alpha\right). \tag{A1}$$

The teacher model is updated via EMA after every student weight update. This scheduling strategy prevents early student errors from being overly preserved in the teacher's targets, which can happen when using a fixed large decay from the very beginning of training.

## A2  ADDITIONAL EXPERIMENTAL RESULTS

Table A1: Ablation study on the EMA decay rate.

| Decay Rate ($\alpha$) | AP | $AP_{50}$ | $AP_{25}$ |
|---|---|---|---|
| 0.99 | **57.3** | **75.3** | **84.3** |
| 0.95 | 55.6 | 75.0 | 83.5 |
| 0.90 | 55.8 | 74.8 | 83.7 |

**Ablation On EMA Decay Rate**  Table A1 presents an ablation study comparing different EMA decay values on the ScanNetV2 validation set. We employ a decay-rate scheduler that gradually increases the decay from a small initial value to the target $\alpha$. This progressive increase allows the teacher to quickly adapt to improvements of the student in early training, while gradually shifting toward a more stable, slowly changing teacher as training progresses. As a result, the pseudo-labels become progressively more stable and provide increasingly reliable supervision. We observe that a

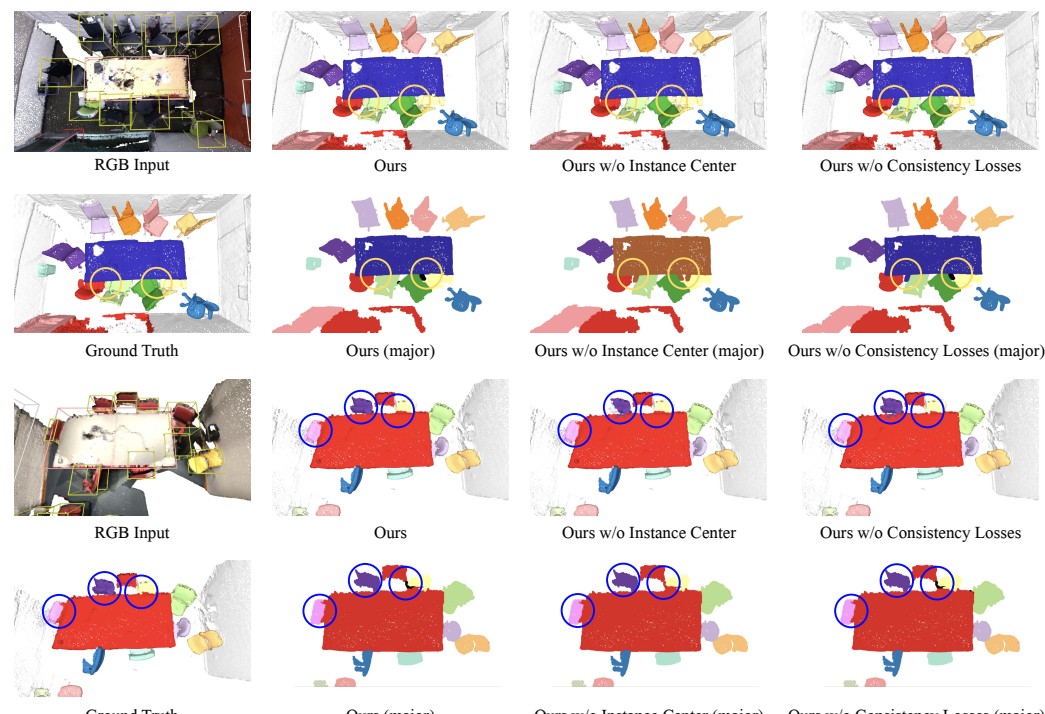

Figure A1: **Qualitative comparison of model variants from the ablation study on ScanNetV2.** Yellow boxes indicate ground-truth instance boxes, and circles highlight overlapping regions. Each row includes the RGB input, ablated model predictions, the ground truth, and an additional visualization highlighting only the predicted mask with the highest IoU(major) for each ground-truth instance.

Table A2: Ablation study on the random seed.

| Seed | AP | $AP_{50}$ | $AP_{25}$ |
|---|---|---|---|
| 1999 (Ours) | 57.3 | **75.3** | 84.3 |
| 100 | 56.5 | 74.7 | 82.5 |
| 200 | **57.5** | 74.7 | 83.2 |
| 300 | 57.1 | 75.2 | **84.7** |
| 400 | 57.3 | 75.1 | 83.3 |
| Mean $\pm$ Std | $57.2 \pm 0.4$ | $75.0 \pm 0.4$ | $83.6 \pm 0.9$ |

final decay of 0.99 yields the strongest performance, and therefore adopt $\alpha = 0.99$ as the default setting in BEEP3D.

**Ablation On Random Seed**    Table A2 reports the performance of BEEP3D trained with different random seeds on the ScanNetV2 validation set. Across five independent runs, the results remain tightly clustered, demonstrating that BEEP3D exhibits stable behavior and that the reported improvements are statistically significant rather than due to random variation. This also confirms that our framework yields reproducible performance under standard training conditions.

**Comparison Under Sketchy-3DIS Noisy-Box Protocols**    Table A3 reports the performance of BEEP3D under the noisy-box protocols defined in Sketchy-3DIS (Deng et al., 2025) on the Scan-NetV2 validation set. Under the $S_1$ setting, which introduces only scaling perturbations, BEEP3D retains strong robustness and even surpasses Sketchy-3DIS in $AP_{50}$, demonstrating that our refinement mechanism is insensitive to scale noise. In contrast, because instance center–based refinement relies directly on the box center coordinates, BEEP3D exhibits a larger performance drop under the $S_2$ setting where centers are additionally perturbed.

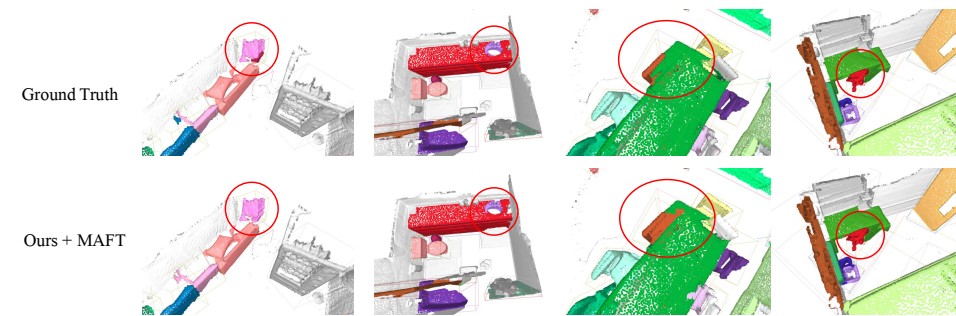

Figure A2: **Failure case on the ScanNetV2 validation set.** One bounding box is fully contained within another. Yellow boxes denote ground-truth instance boxes, and red circles highlight the fully overlapping regions.

Table A3: Comparison under Sketchy-3DIS noisy-box protocols. $S_1$ introduces scale perturbations to each bounding box, and $S_2$ introduces both scale and center-shift perturbations.

| Method | Sup. | $AP_{50}$ | $AP_{25}$ |
|---|---|---|---|
| Box2Mask (Chibane et al., 2022) | | 52.4 | 67.5 |
| GaPro (Ngo et al., 2023a) + SPFormer (Sun et al., 2023) | $S_1$ | 53.5 | 72.2 |
| Sketchy-3DIS (Deng et al., 2025) | | 65.8 | **83.1** |
| Ours + MAFT (Lai et al., 2023) | | **65.9** | 74.9 |
| Sketchy-3DIS (Deng et al., 2025) | $S_2$ | **63.7** | **82.1** |
| Ours + MAFT (Lai et al., 2023) | | 62.5 | 72.5 |

**Qualitative Comparison**    We also provide a qualitative comparison of model variants from the ablation study on the ScanNetV2 validation set. Figure A1 illustrates that removing key components of our method leads to noticeable errors, especially in overlapping regions. We follow the Box2Mask (Chibane et al., 2022) visualization protocol, where each predicted instance mask is assigned the color of the majority ground-truth label contained within that mask. While this produces visually clean overlays, it may also make differences between predictions and ground truth appear less pronounced. To offer a more transparent and interpretable comparison, Figure A1 further includes alternative visualizations in which, for each ground-truth instance, we display only the predicted mask that achieves the highest IoU coverage. This representation avoids color smoothing effects and more clearly highlights discrepancies between variants.  Without instance center–based query refinement, the model often fails to cover entire instances, leaving some points unassigned. When consistency losses are removed, points in overlapping regions are frequently assigned to incorrect instances. These qualitative results further support that our proposed components contribute to more precise and reliable instance segmentation.

## A3   ANALYSIS OF FAILURE CASES

Figure A2 shows a failure case where one bounding box is fully contained within another. In this extreme overlap, supervision becomes ambiguous and can lead to missing the entire inner instance. This highlights a limitation of our approach under severe overlap and occlusion.

## A4   LLM USAGE

We used a large language model (LLM) solely for light copy-editing (grammar and phrasing) of a subset of sentences. All scientific content—including problem formulation, method design, experiments, analysis, and conclusions—was conceived and written by the authors. No text, equations, code, figures, or results were generated by the LLM. All edits suggested by the LLM were manually reviewed by the authors, who take full responsibility for the final content.

