# OpenReview forum: "BEEP3D: Box-Supervised End-to-End Pseudo-Mask Generation for 3D Instance Segmentation"
_ICLR.cc/2026/Conference — Submitted to ICLR 2026_

### Official Review · Reviewer_WQrj · 2025-10-29

**Soundness:** 3
**Presentation:** 3
**Contribution:** 2
**Rating:** 4
**Confidence:** 4

**Summary:**

This paper studies 3D instance segmentation using only 3D bounding boxes. The authors propose BEEP3D: a single-stage training framework that generates pseudo-masks online under a student-teacher scheme where the teacher is updated via EMA from the student; instance-center–guided positional queries for refinement; and two consistency losses, query consistency and mask-feature consistency, to align representations. The method achieves near fully supervised AP on ScanNetV2 and S3DIS and removes the need for a separately pre-trained pseudo-labeler.

**Strengths:**

* **End-to-end pseudo-mask generation**

  * Integrates the pseudo-labeler into the training loop in a single stage, avoiding extra pretraining and simplifying the pipeline and T′ cost.
  * Provides a clear comparison with two-stage methods in terms of pipeline and parameter freezing.

* **Closeness to fully supervised upper bound**

  * On ScanNetV2, the “% full” relative to the corresponding fully supervised method reaches about 98%. Importance: strong performance at the cost of weak supervision.
  * Maintains competitiveness across AP, AP50, and AP25 thresholds. Importance: robustness under different IoU requirements.
  * Achieves leading AP on S3DIS Area 5 as well. Importance: cross-dataset effectiveness.
  * Works with multiple backbones (MAFT, SPFormer). Importance: compatibility.

**Weaknesses:**

* **Insufficient statistical significance and reproducibility information**

  * Does not report variance across multiple runs, confidence intervals, or significance tests.
  * Random seeds and data split/resampling strategies are not specified.
  * No links to code and model weights or a licensing plan are provided.

* **Confirmation bias and error propagation in the pseudo-label loop**

  * The teacher is obtained via EMA of the student; early student errors may be locked into the target ( $\hat{m}_u \cup m_l$ ).
  * Lacks comparisons to “frozen teacher” or “no teacher” alternatives to bound the loop’s benefit.

* **Limited evidence for generalization and noise robustness**

  * Evaluation is limited to ScanNetV2 and S3DIS, which are similar domains.
  * No systematic degradation curves under box noise, offsets, or scale perturbations (related to the Sketchy-3DIS setting).
  * No stratified analysis of cases with heavy occlusion or dense overlap.
  * The ISBNet variant shows “negligible” gains without analysis of causes.
  * Pseudo-mask quality (mACC) is computed only on the training set; no validation-set measure is reported.

**Questions:**

* **Statistical significance and reproducibility**

  * Will you provide links to the code and model weights, including environment files and training scripts?

* **Confirmation bias and error propagation in the pseudo-label loop**

  * What is the effect of varying the EMA decay {0.90, 0.95, 0.99} and the update frequency on AP and training stability?
  * Can you add and explain ablations with a frozen teacher and with no teacher to quantify the loop’s upper bound and its necessity?

* **Generalization and noise robustness**

  * Can you evaluate on additional domains (e.g., different building styles) and report at least cross-domain validation results?
  * Will you sweep box center offsets, scale perturbations, and size noise, plot degradation curves, and compare to a Sketchy-style perturbation protocol?
  * Can you provide a hard-case analysis with AP and visualizations on subsets that exhibit heavy overlap and strong occlusion?

---

> ### Author Response · Authors · 2025-11-21
> **Response #1**
>
> We sincerely appreciate your insightful comments and thoughtful questions.
>
> **W1-1. Regarding insufficient statistical significance.** Does not report variance across multiple runs, confidence intervals, or significance tests.
>
> **A1.** We agree that reporting variance across multiple runs is important for assessing statistical robustness. To address this, we trained BEEP3D with five different random seeds on ScanNetV2. As shown in the table below, the AP values remain highly consistent, with a mean of 57.2 ± 0.4.
> | Seed | AP   | AP50 | AP25 |
> |------|------|------|------|
> | 1999(Ours)    | 57.3 | 75.3 | 84.3 |
> | 100    | 56.5 | 74.7 | 82.5 |
> | 200   | 57.5 | 74.7 | 83.2 |
> | 300   | 57.1 | 75.2 | 84.7 |
> | 400    | 57.3 | 75.1 | 83.3 |
> | **mean ± std** | **57.2 ± 0.4** | **75.0 ± 0.4** | **83.6 ± 0.9** |
>
>
> **W1-2. Regarding reproducibility information.** Random seeds and data split/resampling strategies are not specified. No links to code and model weights or a licensing plan are provided.
>
> **A2.** We appreciate the reviewer’s concern regarding reproducibility. The supplementary material already includes our source code, and the accompanying README provides links to download the model weights used in our experiments. The training seed configurations are specified in the YAML files under the configs directory. We plan to release the full codebase on GitHub, along with model weights and licensing information, upon acceptance of the paper.
>
> **W2-1. Regarding confirmation bias and error propagation in the pseudo-label loop.** The teacher is obtained via EMA of the student; early student errors may be locked into the target.
>
> **A3.** We appreciate the reviewer’s concern and acknowledge that the original manuscript did not sufficiently describe the EMA update mechanism. Importantly, we do not use a fixed decay rate of 0.99 from the beginning. Instead, we adopt a scheduler where the decay rate starts from a lower value and gradually increases toward 0.99 over the course of training. This design ensures that, in the early stages, the teacher model remains highly responsive to updates from the student, preventing early student errors from being locked into the teacher’s predictions. We have clarified this mechanism in the Appendix A1 of the revised manuscript.
>
> **W2-2. Regarding additional ablation studies.** Lacks comparisons to “frozen teacher” or “no teacher” alternatives to bound the loop’s benefit.
>
> **A4.** We appreciate the reviewer’s suggestion to include alternative baselines. A “frozen teacher” setup would require a pretrained teacher model, which contradicts one of the core advantages of BEEP3D—namely, that it avoids any additional pretraining or multi-stage procedures. For the “no teacher” alternative, we have included a direct self-training baseline (refer to A8 in the response to Reviewer UCvp). Removing the teacher also eliminates our Instance center–based query refinement(3.3) in the teacher branch, and consequently disables the teacher–student consistency loss(3.4), leading to further performance degradation (see Table 4 for detailed ablations). The results show a clear performance drop compared to our EMA teacher–student framework, demonstrating that the teacher plays an essential role in stabilizing pseudo-label quality and preventing error amplification. These findings highlight the necessity and effectiveness of the teacher–student framework.
>
> | Method              | AP   | AP50 | AP25 |
> |---------------------|------|------|------|
> | **w/ teacher (Ours)** | **57.3** | **75.3** | **84.3** |
> | w/o teacher         | 53.4 | 72.9 | 82.8 |

---

> ### Author Response · Authors · 2025-11-21
> **Response #2**
>
> **W3-1. Regarding Limited evidence for generalization.** Evaluation is limited to ScanNetV2 and S3DIS, which are similar domains.
>
> **A5.** We agree that broader evaluation would further strengthen our generalization claims.
>  ScanNetV2 and S3DIS are the standard benchmarks used in prior box-supervised 3DIS research, and we followed this setting to ensure fair comparison with existing methods.
>
> To additionally examine performance in a more challenging domain, we evaluated BEEP3D on ScanNet++, which contains significantly more complex geometry and a large label space(~100 classes). As reported in the ScanNet++ benchmark, even fully supervised instance segmentation methods exhibit considerably lower AP on this dataset, reflecting the difficulty of the task. Box-supervised Ours+MAFT achieves 15.4 AP50, falling within the expected range under this challenging setting. Improving box-supervised performance on such fine-grained, high-complexity scenes remains an important open problem and a promising direction for future research.
>
> **Evaluation on Scannet++ Validation set**
> | Method           | AP50 |
> |------------------|:----:|
> | PointGroup | 14.8 |
> | HAIS      | 16.7 |
> | SoftGroup  | 23.7 |
> | Ours + MAFT | 15.4 |
>
> **W3-2. Regarding Limited evidence for noise robustness.** No systematic degradation curves under box noise, offsets, or scale perturbations (related to the Sketchy-3DIS setting).
>
> **A6.** Please refer to A2 of the global response for detailed results. As shown in the additional experiments following the reviewer’s suggestion, BEEP3D achieves the highest AP50 and the second-best AP25 under the S1 noisy-box setting. These findings demonstrate that BEEP3D maintains strong robustness to noisy box annotations.
>
> **W3-3. Regarding failure case analysis.** No stratified analysis of cases with heavy occlusion or dense overlap.
>
> **A7.** As noted in A3 of the global response, we include additional failure-case visualizations in the Appendix A3 of the revised manuscript. In particular, we show examples where BEEP3D fails to separate two instances whose bounding boxes fully overlap in highly cluttered scenes. These cases highlight an inherent limitation of coarse box supervision under heavy occlusion. We believe this provides useful insight into the model’s current limitations and also serves as a strong motivation for future work aimed at addressing such challenging scenarios.
>
> **W3-4. Regarding the ISBNet variant.** The ISBNet variant shows “negligible” gains without analysis of causes.
>
> **A8.** As discussed in A1 of the global response, the limited improvement of BEEP3D when combined with ISBNet is due to the architectural mismatch: our proposed query refinement and consistency losses are specifically designed for transformer-based query–decoder architectures, whereas ISBNet does not employ such mechanisms. This is an architectural compatibility issue rather than a limitation of the proposed method itself.
>
> **W3-5. Regarding pseudo-mask quality on the validation set.** Pseudo-mask quality (mACC) is computed only on the training set; no validation-set measure is reported.
>
> **A9.** We evaluated pseudo-mask quality on the training set to ensure a fair comparison with BSNet, which also reports mACC only during training. In addition, we computed pseudo-mask quality in the overlapping regions on the validation set, obtaining an mACC of 77.0%, which is comparable to the mACC measured on the training set.

---

> ### Author Response · Authors · 2025-11-21
> **Response #3**
>
> **Q1. Regarding Statistical significance and reproducibility.** Will you provide links to the code and model weights, including environment files and training scripts?
>
> **A10.** All materials necessary for reproducibility—including the source code, model weights, configuration files, and training scripts—are provided in the supplementary materials.
>
> **Q2-1. Regarding Confirmation bias and error propagation in the pseudo-label loop.** What is the effect of varying the EMA decay {0.90, 0.95, 0.99} and the update frequency on AP and training stability?
>
> **A11.** We provide an ablation study on different EMA decay rates in the table below. The teacher network is updated after every parameter update of the student model. These results illustrate how varying the decay factor influences performance and training stability:
>
> | EMA decay (α) | AP | AP50 | AP25 |
> |---:|---:|---:|---:|
> | 0.99 | **57.3** | **75.3** | **84.3** |
> | 0.95 | 55.6 | 75.0 | 83.5 |
> | 0.90 | 55.8 | 74.8 | 83.7 |
>
> These results verify that a decay rate of 0.99 provides the best overall performance and stability.
>
> **Q2-2. Regarding Confirmation bias and error propagation in the pseudo-label loop.** Can you add and explain ablations with a frozen teacher and with no teacher to quantify the loop’s upper bound and its necessity?
>
> **A12.** Please see the answer A4 above.
>
> **Q3-1. Regarding generalizability.** Can you evaluate on additional domains (e.g., different building styles) and report at least cross-domain validation results?
>
> **A13.** Please see the answer A5 above.
>
> **Q3-2. Regarding sketchy style perturbation protocol.** Will you sweep box center offsets, scale perturbations, and size noise, plot degradation curves, and compare to a Sketchy-style perturbation protocol?
>
> **A14.** Please see the answer A6 above.
>
> **Q3-3. Regarding hard-case analysis.** Can you provide a hard-case analysis with AP and visualizations on subsets that exhibit heavy overlap and strong occlusion?
>
> **A15.** Please see the answer A7 above.

---

> > ### Comment · Reviewer_WQrj · 2025-11-27
> >
> > Thank you for the detailed rebuttal and for adding several new experiments. I keep my current score for now, but I am willing to raise it if you can clarify two remaining points:
> >
> > 1. I understand that you avoid a frozen teacher in order to stay within the “box only, single stage” setting. However, I am still not fully convinced about the benefit of the EMA teacher itself. Conceptually, I am very curious about the following question. If you use a fully trained network as a frozen teacher to generate pseudo labels once, and then train the student on these pseudo labels, would this achieve better or comparable performance than your EMA teacher that keeps updating during training. In other words, where exactly does the performance gain of the EMA update come from, compared with using a strong fixed teacher or even using the student to self-train without EMA in 3D instance segmentation. If you have any empirical comparison or more concrete justification here, it would help a lot.
> >
> > 2. For the ISBNet variant, I find your current explanation not fully satisfactory. At the moment the answer mainly attributes the small or negative gain to “architectural mismatch”, which feels too high level. I agree that your modules are naturally aligned with transformer style query–decoder backbones, and I am fine if the paper explicitly scopes the main claims to that family. However, for ISBNet I would still expect a more concrete analysis. For example, can you show how pseudo mask quality or error modes change on ISBNet, which component fails first, or whether reasonable adaptations or hyperparameter tuning can recover part of the gain. Without such analysis, the “mismatch” explanation reads more like a post hoc justification rather than an observed failure mode.
> >
> > If you can address these two points in a more concrete way, I am happy to reconsider my score in a positive direction.

---

> ### Author Response · Authors · 2025-12-02
> **Response on additional comment**
>
> We appreciate the reviewer’s additional comments and positive feedback. The constructive suggestions are highly valuable for further improving our work.
>
> **Q1. Regarding the frozen teacher and the benefit of the EMA teacher.**
>
> **A1.** We first clarify that a “fully trained” box-supervised teacher does not necessarily provide the best pseudo-labels. Prior box-supervised methods typically rely on separate frozen pseudo-labelers with additional information (e.g., geometric priors, probabilistic assumptions, or simulated data). In contrast, our results (Table 4) show that an EMA teacher in BEEP3D produces better pseudo-masks than existing fully trained box-supervised teachers such as GaPro or BSNet.
> The key reason is that a frozen teacher generates pseudo-labels that never improve: its errors in overlapping regions are fixed, so a student trained on these labels cannot exceed the teacher’s own limitations.
> EMA avoids this limitation. Because box supervision provides accurate labels on non-overlapping regions, the student receives a stable and reliable learning signal early on. As the student improves its predictions in ambiguous regions, the EMA teacher gradually incorporates these improvements, yielding progressively higher-quality pseudo-labels. This co-evolution is not possible with a frozen teacher.
> In summary, EMA does not provide gains by itself, but enables the pseudo-labeler to evolve beyond what any frozen box-supervised teacher can provide—leading to stronger masks than fully trained frozen teachers and enabling our single-stage training pipeline.
>
> **Q2. Regarding the ISBNet variant.**
>
> **A2.** We agree that a more concrete explanation is needed for the limited improvement on ISBNet. The key issue is that ISBNet does not maintain global instance queries; instead, it forms instance candidates and aggregates only local point neighborhoods through ball-query operations. Our Instance Center-based Query Refinement assumes a global query mechanism that can re-attend to the entire point cloud after refinement, which is how it corrects ambiguous or overlapping regions in Transformer-based decoders. When applied to ISBNet, this global interaction is replaced by a strictly local aggregation pipeline, so the refinement signal cannot propagate beyond each local cluster. As a result, the refinement module becomes the first component to lose effectiveness, and its contribution is largely suppressed by ISBNet’s architectural design rather than by hyperparameters.

---

### Official Review · Reviewer_bt6s · 2025-10-30

**Soundness:** 2
**Presentation:** 3
**Contribution:** 2
**Rating:** 4
**Confidence:** 5

**Summary:**

The paper tackles the challenge of 3D instance segmentation under weak supervision, specifically from 3D bounding boxes instead of dense point-level annotations. The authors propose BEEP3D, a student-teacher framework where the teacher generates pseudo-masks that guide the student network in an end-to-end manner. Unlike prior methods that either require pretraining a separate pseudo-labeler or rely on heavy geometric priors, BEEP3D integrates pseudo-mask generation directly into training. It uses instance center–based query refinement instead of employing learnable parameters. Two consistency losses are introduced to align the student and teacher representations. The framework is implemented on MAFT, ISBNet, and  SPFormer models, and tested on the ScanNetV2 and S3DIS benchmarks, where it showed competitive results compared to fully supervised methods. The method also cuts training complexity compared to previous weakly supervised methods.

**Strengths:**

- The method achieves near-supervised accuracy on two standard datasets.
- The writing is mostly clear, the figures are illustrative, and the ablation studies are well-structured. Visual results show improved segmentation in overlapping regions.
- Unlike BSNet or GaPro, BEEP3D eliminates an extra pseudo-labeling stage, reducing training time without sacrificing accuracy.

**Weaknesses:**

- The key ideas proposed in this paper (EMA teacher-student updates, query refinement, and consistency-based losses) are well-established concepts adapted from semi-supervised and weakly supervised learning to 3DIS.
- The authors note that when integrated with ISBNet (a non-transformer-based network), BEEP3D underperforms compared to BSNet on S3DIS. This suggests that the framework’s advantages rely heavily on transformer-based query mechanisms, limiting applicability to other architectures.
- While the paper explicitly positions BEEP3D as transformer-based, it tests only on MAFT and SPFormer. Since there are several other transformer architectures for 3DIS (e.g., Mask3D, OneFormer3D, QueryFormer), broader evaluation across more transformer variants would strengthen claims of generality even within its intended paradigm.

**Questions:**

- Can the authors analyze pseudo-mask quality over training epochs to substantiate claims about refinement?
- Can the authors evaluate their framework across a wider set of transformer-based 3DIS methods (1 or 2 extra) to strengthen their evaluation on the two chosen datasets?
- There seems to be a minor issue with the citation format throughout the paper.

---

> ### Author Response · Authors · 2025-11-21
> **Response #1**
>
> We appreciate your thoughtful comments and supportive reviews.
>
> **W1. Regarding key concepts of BEEP3D.** The key ideas proposed in this paper (EMA teacher-student updates, query refinement, and consistency-based losses) are well-established concepts adapted from semi-supervised and weakly supervised learning to 3DIS.
>
> **A1.** We acknowledge the reviewer’s observation that EMA-based teacher–student frameworks are commonly used in semi-supervised learning, and that box-supervised 3DIS methods have also explored variants of this idea. However, the contribution of BEEP3D lies not in reusing EMA itself, but in designing new components that make the teacher–student framework effective under box-level supervision within a single-stage training paradigm.
> Specifically, we introduce Instance Center-based Query Refinement and two consistency losses that are tailored for coarse box annotations and enable the framework to generate high-quality masks without additional supervision or multi-stage training. These components substantially improve performance while maintaining training efficiency, which we believe constitutes the key novelty of BEEP3D beyond the general EMA framework.
>
> **W2. Regarding reliance on transformer-based architectures.** The authors note that when integrated with ISBNet (a non-transformer-based network), BEEP3D underperforms compared to BSNet on S3DIS. This suggests that the framework’s advantages rely heavily on transformer-based query mechanisms, limiting applicability to other architectures.
>
> **A2.** Please see A1 of the global response for a detailed explanation. As discussed there, the limited improvement on ISBNet arises from a structural mismatch: ISBNet replaces global instance queries with local candidate-based aggregation, preventing our query refinement and consistency losses from operating as intended. These components are designed for global query–decoder mechanisms, and thus naturally integrate with transformer-based architectures. This does not indicate a fundamental weakness, but rather that BEEP3D is tailored for the modern transformer-based paradigm.
>
> **W3. Regarding other base models.** While the paper explicitly positions BEEP3D as transformer-based, it tests only on MAFT and SPFormer. Since there are several other transformer architectures for 3DIS (e.g., Mask3D, OneFormer3D, QueryFormer), broader evaluation across more transformer variants would strengthen claims of generality even within its intended paradigm.
>
> **A3.** We thank the reviewer for this suggestion. To further validate generality within the transformer-based family, we additionally evaluate BEEP3D on OneFormer3D. As shown in the A1 of the global response, BEEP3D achieves 94.1% of the fully supervised AP on this architecture, consistent with our results on MAFT and SPFormer. This confirms that BEEP3D generalizes robustly across multiple transformer variants within its intended design scope.
>
> **Q1. Regarding pseudo-mask quality.** Can the authors analyze pseudo-mask quality over training epochs to substantiate claims about refinement?
>
> **A4.** To evaluate how pseudo-mask quality evolves during training, we measured the mean Accuracy (mACC) at different training progress stages (20%, 40%, 60%, 80%, 100%). As shown below, the mACC steadily increases as training proceeds, and stabilizes at a high level after approximately 60% of training:
> | Training Progress (%) | 20   | 40   | 60   | 80   | 100  |
> |-----------------------|-----:|-----:|-----:|-----:|-----:|
> | mACC (%)              | 41.5 | 61.4 | 74.0 | 76.1 | 76.3 |
>
> These results corroborate that the teacher-student updates and query refinement progressively improve pseudo-mask quality throughout training. Here, training progress (%) denotes the proportion of total training epochs.
>
> **Q2. Regarding other base models.** Can the authors evaluate their framework across a wider set of transformer-based 3DIS methods (1 or 2 extra) to strengthen their evaluation on the two chosen datasets?
>
> **A5.** Please see the answer A3 above.
>
> **Q3. Regarding format issues.** There seems to be a minor issue with the citation format throughout the paper.
>
> **A6.** We appreciate the reviewer’s attentive feedback. We have corrected the citation formatting issues in the revised manuscript.

---

### Official Review · Reviewer_v8Bp · 2025-10-31

**Soundness:** 3
**Presentation:** 3
**Contribution:** 3
**Rating:** 8
**Confidence:** 4

**Summary:**

This paper introduces BEEP3D, a novel framework for 3D instance segmentation using only 3D bounding box supervision. The core challenge is the ambiguity from overlapping boxes. BEEP3D cleverly solves this with an end-to-end student-teacher framework, where the teacher model acts as a pseudo-labeler and is updated via an EMA of the student. This unifies pseudo-mask generation and segmentor training into a single stage.

**Strengths:**

1. The end-to-end, single-stage paradigm is a significant advancement over prior multi-stage methods (e.g., GaPro, BSNet). It's elegant, efficient, and avoids reliance on simulated data or complex priors.

2. The instance center-based query refinement is a smart strategy to leverage the most reliable signal from the weak supervision. The dual consistency losses provide robust supervision for the student model.

3. The method achieves state-of-the-art (or highly competitive) performance on ScanNetV2 and S3DIS. Impressively, it closes the gap to fully-supervised methods, achieving 98.1% of the full-supervision AP on the ScanNetV2 validation set.

4. As shown in Table 5, the method eliminates the separate pre-training time ($T'$) for pseudo-label generation, making it efficient and practical.

Clarity: The paper is well-written, and the method is presented clearly.

**Weaknesses:**

1. Evaluation on ScanNet++: It would be valuable to include results on ScanNet++, which offers more challenging indoor scenes and richer geometric details. This would further demonstrate the generalization ability of the proposed method.


2. Failure Case Discussion: Adding a brief discussion of failure cases (e.g., in extremely cluttered scenes) would provide a more complete picture of the method's limitations and guide future work.

**Questions:**

See weaknesses above.

---

> ### Author Response · Authors · 2025-11-21
> **Response #1**
>
> We appreciate the reviewer’s positive evaluation of our work, particularly their recognition of the effectiveness of our query refinement strategies and the efficiency enabled by our single-stage training paradigm.
>
> **W1. Regarding additional evaluation on ScanNet++.** Evaluation on ScanNet++: It would be valuable to include results on ScanNet++, which offers more challenging indoor scenes and richer geometric details. This would further demonstrate the generalization ability of the proposed method.
>
> **A1.** We evaluated Ours + MAFT under the box-supervised setting on the ScanNet++ validation set. The fully supervised results included in the table below are the official validation scores reported in the ScanNet++ paper [1]. Most existing box-supervised 3DIS methods have only been evaluated on ScanNetV2 and do not report results on ScanNet++, so direct comparison with prior box-supervised baselines is unfortunately not possible at this time.
>
> ScanNet++ introduces a large label space (~100 classes) and exhibits significantly more complex geometry than ScanNetV2, making it a highly challenging benchmark where even fully supervised methods achieve relatively low AP. Within this context, our results provide an initial indicator of the limitations and potential of box-supervision in large-scale, fine-grained indoor scenes. We will clarify that improving performance in such settings remains an important open problem and a promising direction for future work.
>
> **Evaluation on Scannet++ Validation set**
>
> | Method           | AP50 |
> |------------------|:----:|
> | PointGroup | 14.8 |
> | HAIS      | 16.7 |
> | SoftGroup  | 23.7 |
> | Ours + MAFT | 15.4 |
>
> [1] Yeshwanth, Chandan, et al. "Scannet++: A high-fidelity dataset of 3d indoor scenes." Proceedings of the IEEE/CVF International Conference on Computer Vision. 2023.
>
> **W2. Regarding failure case analysis.** Failure Case Discussion: Adding a brief discussion of failure cases (e.g., in extremely cluttered scenes) would provide a more complete picture of the method's limitations and guide future work.
>
> **A2.** As mentioned in A3 of the global response, we include additional failure-case visualizations and analysis in the Appendix A3 of the revised manuscript. Consistent with the reviewer’s observation, BEEP3D primarily fails in highly cluttered scenes where the bounding box of one instance heavily overlaps with that of another. These scenarios highlight the inherent limitations of coarse box supervision, and we hope that the added discussion provides clearer insight into these challenging failure modes and directions for future improvement.

---

### Official Review · Reviewer_UCvp · 2025-11-01

**Soundness:** 2
**Presentation:** 2
**Contribution:** 2
**Rating:** 4
**Confidence:** 5

**Summary:**

The authors propose BEEP3D, an end-to-end framework for weakly supervised 3D instance segmentation using only 3D bounding box annotations. The method employs a student-teacher architecture where the teacher model is updated via EMA of the student model and serves as a pseudo-labeler. To improve pseudo-mask quality in ambiguous overlapping regions, the framework introduces two key techniques: (1) instance center-based query refinement that leverages center coordinates from bounding boxes as strong priors to guide the teacher model, and (2) two consistency losses (query consistency and masked-feature consistency) to ensure alignment between student and teacher models at both query and feature representation levels.

**Strengths:**

1.	BEEP3D integrates pseudo-label generation and segmentation model training within a unified training loop. This eliminates the need for separate pre-training stages, significantly simplifying the training pipeline and improving overall efficiency.
2.	The method exploits strong geometric priors implicit in box annotations, specifically instance center coordinates. By enforcing the teacher model's position queries to consistently aggregate these center points, it provides robust spatial guidance for the model.
3.	To ensure effective knowledge transfer from teacher to student, the paper designs two novel consistency losses.

**Weaknesses:**

1.	Experimentally, on the validation set, the method achieves relatively limited AP improvements. More critically, it underperforms BSNet+MAFT across both AP₅₀ and AP₂₅ metrics. On the test set, the absence of corresponding performance data from competing methods prevents fair comparison. Additionally, comparisons with the latest state-of-the-art methods are missing.
2.	The method's core innovations, particularly Instance Center-based Query Refinement and Query Consistency Loss, heavily rely on Transformer-specific query mechanisms, making them relatively incremental. It is not a general weakly supervised approach, when applied to non-Transformer architectures, performance even degrades, as shown in Table 2 where Ours + ISBNet underperforms BSNet + ISBNet.
3.	The Instance Center-based Query Refinement critically depends on accurate instance center points extracted from bounding boxes. The design does not address method robustness when box annotations contain noise (common in weak supervision). Meanwhile, the hard arg max assignment in pseudo-mask generation may lead to rapid performance degradation when teacher model predictions are inaccurate (error accumulation).

**Questions:**

1.	Could the authors provide error analysis to better understand failure cases and model limitations?
2.	In Table 1 validation set, although BEEP3D's AP (57.3) is slightly higher than BSNet (56.2), it shows comprehensive deterioration in both AP₅₀ and AP₂₅ metrics. Does this indicate that BEEP3D sacrifices instance detection recall to optimize high-IoU segmentation precision? Can the authors explain the underlying causes of this critical metric regression?
3.	Given that student and teacher models share identical network architecture, a more straightforward self-training baseline seems viable: using only a student model where predictions at step N generate pseudo-masks to supervise training at step N+1. Can the authors justify the necessity of adopting the student-teacher framework compared to this more direct self-training loop?

---

> ### Author Response · Authors · 2025-11-21
> **Response #1**
>
> We appreciate your detailed review and constructive feedback.
>
> **W1-1. Regarding limited performance improvements.** Experimentally, on the validation set, the method achieves relatively limited AP improvements. More critically, it underperforms BSNet+MAFT across both AP₅₀ and AP₂₅ metrics.
>
> **A1.** We would like to clarify that an improvement of +1.1 AP in 3D instance segmentation is far from negligible. AP(50–95) is particularly difficult to improve because it averages over high IoU thresholds (up to 0.95), which require consistently more precise instance boundaries. Even fully supervised methods typically report improvements in the range of +0.7 to +2.1 AP (see Table 1), further indicating the challenge of achieving such gains.
> In the box-supervised setting, this improvement becomes even more meaningful. Box annotations provide only coarse localization, making high-IoU mask refinement substantially harder than in fully supervised setups. Achieving +1.1 AP under such weak supervision demonstrates that BEEP3D effectively produces high-quality, fine-grained instance masks that go beyond what coarse box inputs can provide.
> Therefore, we believe that the reported AP improvement represents a significant and meaningful gain for 3D instance segmentation, and especially within the more challenging box-supervised paradigm.
> Regarding AP50 and AP25, these metrics primarily measure coarse localization, which is largely saturated under box-level supervision. Our method improves high-IoU mask quality through instance-center query refinement and consistency losses, which is why the gains primarily appear in AP(50–95).
>
> **W1-2. Regarding test set comparisons and missing SOTA baselines.** On the test set, the absence of corresponding performance data from competing methods prevents fair comparison. Additionally, comparisons with the latest state-of-the-art methods are missing.
>
> **A2.** We agree that fair test-set comparison is important. However, for ScanNetV2, the benchmark policy prohibits reproducing test-set results for methods that did not officially submit their predictions. As a result, if prior works did not report their test performance, it is not possible for us to reproduce or infer them. In our paper, we therefore compared against all publicly available test-set results from existing box-supervised 3DIS methods.
> To the best of our knowledge, the most recent box-supervised 3DIS methods are Sketchy-3DIS and BSNet, both of which we included in our comparisons. If any additional SOTA baselines were missed, we are willing to incorporate them during the rebuttal period.
>
> **W2. Regarding reliance on transformer-based architectures.** The method's core innovations, particularly Instance Center-based Query Refinement and Query Consistency Loss, heavily rely on Transformer-specific query mechanisms, making them relatively incremental. It is not a general weakly supervised approach, when applied to non-Transformer architectures, performance even degrades, as shown in Table 2 where Ours + ISBNet underperforms BSNet + ISBNet.
>
> **A3.** As discussed in A1 of the global response, the limited improvement on ISBNet arises from an architectural mismatch: our Instance Center-based Query Refinement and Consistency Losses are specifically designed for transformer-based query–decoder architectures. This is not a limitation of the proposed method itself but rather a consequence of ISBNet not employing such mechanisms. Please refer to A1 in the global response for the full explanation.
>
> **W3-1. Regarding noisy annotation.** The Instance Center-based Query Refinement critically depends on accurate instance center points extracted from bounding boxes. The design does not address method robustness when box annotations contain noise (common in weak supervision).
>
> **A4.**  Please refer to A2 of the global response for a detailed explanation. As summarized there, when training under the S1 noisy-box setting defined in Sketchy-3DIS, BEEP3D achieves the best AP50, indicating stronger robustness to noisy annotations compared to typical box-supervised methods. However, BEEP3D shows a gap in AP25 relative to Sketchy-3DIS, suggesting that there remains room for improving coarse localization under severe box noise.

---

> ### Author Response · Authors · 2025-11-21
> **Response #2**
>
> **W3-2. Regarding hard argmax assignment.** Meanwhile, the hard arg max assignment in pseudo-mask generation may lead to rapid performance degradation when teacher model predictions are inaccurate (error accumulation).
>
> **A5.** The hard arg-max assignment is used to ensure a unique and unambiguous allocation of points to one instance in overlapping regions. Importantly, this step does not cause rapid performance degradation because the student is supervised through a structured combination of:
>  (1) deterministic labels from non-overlapping regions, and
>  (2) teacher-generated pseudo-masks for the remaining areas.
>  Even though the teacher model predictions may contain noise, this supervision structure constrains the impact of ambiguous overlapping regions and prevents uncontrolled error propagation.
> Moreover, in the EMA update of the teacher, the decay rate gradually increases toward 0.99 during training. Early in training—when the decay rate is low—the teacher remains highly responsive to updates from the student, allowing early errors to be corrected rather than accumulated. This progressive adjustment plays a key role in preventing confirmation-bias amplification. We describe this mechanism in more detail in the revised manuscript.
>
>
> **Q1. Regarding failure cases analysis.** Could the authors provide error analysis to better understand failure cases and model limitations?
>
> **A6.** Please see A3 of the global response for a detailed discussion. In summary, BEEP3D inherits the inherent limitations of box supervision and tends to fail in highly cluttered scenes where the bounding boxes of two instances fully overlap. In such cases, the coarse box annotations provide insufficient cues for correct instance separation.
>
> **Q2. Regarding metric regression.** In Table 1 validation set, although BEEP3D's AP (57.3) is slightly higher than BSNet (56.2), it shows comprehensive deterioration in both AP₅₀ and AP₂₅ metrics. Does this indicate that BEEP3D sacrifices instance detection recall to optimize high-IoU segmentation precision? Can the authors explain the underlying causes of this critical metric regression?
>
> **A7.** As discussed in A1, we clarify that the improvement of +1.1 AP is a notable and meaningful gain in 3DIS. AP50 and AP25 primarily measure coarse localization, and because box supervision inherently provides only coarse spatial information, box-supervised methods tend to perform well on these low-IoU metrics. This can be seen in Table 1, where BSNet+MAFT even matches or outperforms the fully supervised MAFT on AP₅₀ (75.9 vs. 75.9) and AP₂₅ (85.7 vs. 84.5), despite relying solely on box annotations.
> In contrast, the improvements brought by BEEP3D mainly stem from query refinement and consistency losses, which emphasize mask precision rather than coarse recall. These components naturally boost performance in the high-IoU regime captured by AP(50–95), while offering limited impact on coarse-IoU thresholds such as AP50 and AP25.
>
> **Q3. Regarding the self-training loop.** Given that student and teacher models share identical network architecture, a more straightforward self-training baseline seems viable: using only a student model where predictions at step N generate pseudo-masks to supervise training at step N+1. Can the authors justify the necessity of adopting the student-teacher framework compared to this more direct self-training loop?
>
> **A8.** We provide an additional experiment comparing BEEP3D with a direct self-training baseline, where a single model generates pseudo-masks at step N to supervise training at step N+1. The results show that this baseline performs noticeably worse than our default teacher–student framework with EMA updates. Moreover, removing the teacher also eliminates our Instance center–based query refinement(3.3) in the teacher branch, and consequently disables the teacher–student consistency loss(3.4), leading to further performance degradation (see Table 4 for detailed ablations). This confirms that the EMA teacher plays an important stabilizing role by smoothing noisy student predictions, reducing error oscillation, and providing more consistent pseudo-labels throughout training.
> | Method              | AP   | AP50 | AP25 |
> |---------------------|------|------|------|
> | **w/ teacher (Ours)** | **57.3** | **75.3** | **84.3** |
> | w/o teacher         | 53.4 | 72.9 | 82.8 |

---

> > ### Comment · Reviewer_UCvp · 2025-11-26
> > **Follow up on author's rebuttal**
> >
> > I thank the authors for their response and the additional self-training experiment. However, I still have several critical concerns regarding the effectiveness of performance improvement and the scope of application of methods.
> >
> > **1. Insufficient statistical significance and Metric regression (RE: A1&A7)**
> >
> > The author reported a weak improvement (against from a paper in Mar. 2024) in the validation set+1.1 AP. Due to the lack of reporting on the mean and standard deviation of multiple experimental runs in the paper, it is currently uncertain whether this improvement is a genuine improvement in the method or solely due to random variance. Furthermore, I cannot agree that the regression of $AP_{50}$ (from 75.9 to 75.3) can be ignored (the authors argue that coarse metrics are "saturated"). A robust SOTA method should ideally improve (or at least maintain) detection recall. The weak and lack of statistical validation of AP improvement, accompanied by a clear $AP_ {50} $ reduction, indicates that this method has an adverse performance trade-off.
> >
> > **2.	Limited generality (Regarding response A3)**
> >
> > The author acknowledges that this method is incompatible with non Transformer architectures such as ISBNet. This confirms my concern that the work is more like a customized design for a specific Query based architecture, rather than a general weakly supervised instance segmentation framework, which limits its widespread impact.
> >
> > **3.	Dependence on ideal priors (Regarding Reply A4)**
> >
> > The performance of this method is highly dependent on "instance center query refinement" (contributing+3.4 AP in Table 4), which assumes that we can extract accurate instance centers from bounding boxes. Specifically, no experiments were conducted where noise/perturbation is added to the Instance Centers (simulating inaccurate centers from noisy boxes).The lack of robustness mechanisms for noisy bounding boxes (which are common in weak supervision) remains a significant limitation.
> >
> > **4.	Concerns on Visualization (Re: Figure A1)**
> >
> > The qualitative results provided in the paper are not comprehensive. Some of the figures and tables in the paper look a bit rushed. For example in Figure A1, the "Ours" predictions appear almost identical to the "Ground Truth". This level of pixel-perfect alignment in a weakly supervised setting and raises concerns about the representative nature of the selected visualizations (potential Cherry-picking).
> >
> > I will maintain my original score and think that this paper did not meet the bar of ICLR.

---

> > > ### Author Response · Authors · 2025-12-02
> > > **Response on additional comment #1**
> > >
> > > Thank you for the additional comments and efforts. We sincerely appreciate your feedback.
> > >
> > > **Q1. Insufficient statistical significance and Metric regression(RE: A1&A7)**
> > >
> > > **A1.** We thank the reviewer for highlighting this important point. We have now conducted additional multi-seed experiments, and the results confirm that the +1.1 AP improvement is stable rather than a random fluctuation. We will include the full statistics in the revised manuscript to address the reviewer’s concern regarding significance and reproducibility.
> > >
> > > | Seed | AP   | AP50 | AP25 |
> > > |------|------|------|------|
> > > | 1999(Ours)    | 57.3 | 75.3 | 84.3 |
> > > | 100    | 56.5 | 74.7 | 82.5 |
> > > | 200   | 57.5 | 74.7 | 83.2 |
> > > | 300   | 57.1 | 75.2 | 84.7 |
> > > | 400    | 57.3 | 75.1 | 83.3 |
> > > | **mean ± std** | **57.2 ± 0.4** | **75.0 ± 0.4** | **83.6 ± 0.9** |
> > >
> > > Regarding the potential trade-off, we agree that a strong method should ideally preserve detection recall. Despite relying solely on box annotations and a single-stage training paradigm, BEEP3D retains 99.2% and 99.7% of the fully supervised MAFT performance on AP50 and AP25, respectively. This demonstrates that BEEP3D maintains strong recall performance while still providing meaningful improvements in high-IoU mask quality within the box-supervised setting.
> > >
> > > **Q2. Limited generality (Regarding response A3)**
> > >
> > > **A2.**  We appreciate the reviewer’s concern regarding generality beyond a single architecture. As stated in L92 of the manuscript, BEEP3D is explicitly designed for Transformer-based backbones with instance-level queries—a paradigm that now dominates high-performing 3DIS models. To demonstrate that our method is not tied to a single architecture, we additionally evaluated BEEP3D on OneFormer3D, a Transformer-based model distinct from MAFT and SPFormer. As shown below, our framework achieves 94.1% of the fully supervised performance in AP, supporting its applicability across multiple Transformer variants.
> > >
> > > | Sup.   | Method              | AP    | AP50  | AP25  |
> > > |-------|----------------------|-------|-------|-------|
> > > | Fully | OneFormer3D          | 59.3  | 78.1  | 86.4  |
> > > | Box   | OneFormer3D + Ours   | 55.8  | 74.7  | 84.6  |
> > >
> > > Regarding ISBNet, we agree that BEEP3D brings only negligible gains. This is due to a structural mismatch, not a conceptual limitation. ISBNet does not maintain global instance queries; instead, it generates instance candidates and aggregates local point features via ball-query. Our Instance Center-based Query Refinement, however, relies on queries that attend to the entire point cloud(Sec. 3.3). When the global query mechanism is replaced with ISBNet’s local aggregation pipeline, the refinement effect is substantially reduced. This explains the lack of improvement on ISBNet more concretely than our initial high-level statement.
> > > Importantly, across three distinct Transformer architectures (SPFormer, MAFT, and OneFormer3D), BEEP3D consistently achieves over 90% of the fully supervised AP, demonstrating that the method is effective across the Transformer-based architectures that dominate modern 3DIS benchmarks.

---

> ### Author Response · Authors · 2025-12-02
> **Response on additional comment #2**
>
> **Q3. Dependence on ideal priors (Regarding Reply A4)**
>
> **A3.** We thank the reviewer for highlighting this important point. While the S1 protocol of Sketchy-3DIS perturbs box scales without altering instance centers, we have also conducted experiments under the S2 setting, where center locations are perturbed. The results are as follows.
>
> **Evaluation on noisy Bounding-Box Annotations (S2)**
>
> | Method            | AP50 | AP25 |
> |-------------------|:----:|:----:|
> | Sketchy-3DIS      | **63.7** | **82.1** |
> | Ours + MAFT       | 62.5 | 72.5 |
>
> BEEP3D exhibits a larger performance drop under the S2 setting compared to the degradation that Sketchy-3DIS reports under its S1 protocol. At the same time, we would like to clarify the intended scope of BEEP3D. Similar to prior box-supervised 3DIS methods, our framework assumes clean box annotations, as provided in standard benchmarks such as ScanNetV2 and S3DIS. Robustness to substantial centroid perturbations is therefore outside the standard evaluation protocol for box-supervised 3DIS, but remains an interesting and valuable direction for future exploration.
> Importantly, under the S1 noisy-box condition, BEEP3D already demonstrates stronger robustness than prior box-supervised baselines in AP50—including Sketchy-3DIS. This indicates that our refinement mechanism remains stable under realistic scale noise, even though explicit modeling of perturbations is not incorporated.
>
> **Q4. Concerns on Visualization (Re: Figure A1)**
>
> **A4.** We appreciate the reviewer’s concern regarding the qualitative results. We would like to clarify that no cherry-picking was performed when selecting the visualizations. The figures in the original manuscript were generated following the Box2Mask visualization protocol, where each predicted instance mask is colored using the majority ground-truth label within that mask. This often produces visually clean overlays, which may give the impression of near-perfect alignment.
> To provide a more transparent comparison, the revised manuscript includes additional visualizations where we display only the predicted instance mask with the highest IoU coverage for each ground-truth instance. This reveals the actual discrepancies more clearly.
> If the reviewer has further suggestions for improving clarity, we are happy to incorporate them in the revised version.

---

### Author Response · Authors · 2025-11-21
**Global response #1**

Dear reviewers,

We sincerely appreciate thoughtful comments and detailed feedback. We have carefully addressed all the raised concerns in our rebuttal and revised manuscript(**all changes are highlighted in blue**).

First, we deeply appreciate your recognition of the following strengths in our work:
- SOTA performances on ScanNetV2 and S3DIS, closing the gap to fully supervised methods.
- Training efficiency derived from the end-to-end single-stage training paradigm.

Additionally, we are grateful for the opportunity to clarify and address several key concerns raised in the reviews.

**Q1.** Negligible performance improvements on ISBNet and limitations on transformer-based architecture. (Reviewer UCvp, bt6s, WQrj)

**A1.** Transformer decoders with object/instance queries have become a mainstream design paradigm in recent 3DIS, as demonstrated by works such as Mask3D [1] and MAFT [2], and by the fact that many of the top-performing methods on the ScanNetV2 test server [3–6] adopt transformer-based architectures. Our modules—Instance Center-based Query Refinement and Consistency Losses—are therefore intentionally developed within this widely adopted query–decoder framework.
Regarding the limited gains observed with ISBNet, we agree that our components are tightly coupled with transformer-based architectures and are not yet fully optimized for non-transformer backbones. As discussed in the paper, this reflects an architectural mismatch rather than a conceptual limitation of BEEP3D. We view this not as a drawback but as a natural consequence of designing components tailored to the dominant architecture family in modern 3DIS. To further demonstrate BEEP3D’s generality within transformer-based models, we additionally evaluated BEEP3D on OneFormer3D[7].

| Sup.   | Method              | AP    | AP50  | AP25  |
|-------|----------------------|-------|-------|-------|
| Fully | OneFormer3D          | 59.3  | 78.1  | 86.4  |
| Box   | OneFormer3D + Ours   | 55.8  | 74.7  | 84.6  |

From the above results, our framework achieves 94.1% of the fully supervised performance in AP, supporting its applicability across multiple transformer variants.

**Q2.** Robustness on noisy box annotation. (Reviewer UCvp, WQrj)

**A2.** Following the S1 noisy-box protocol of Sketchy-3DIS, we injected noise by scaling each bounding box by 5% along every axis and trained Ours + MAFT under this perturbed supervision. We then evaluated the models on the ScanNetV2 validation set. As shown in the table below, our method achieves the highest AP50 (65.9%), demonstrating strong robustness to noisy box annotations.

**Evaluation on noisy Bounding-Box Annotations (S1)**

| Method            | AP50 | AP25 |
|-------------------|:----:|:----:|
| Box2Mask          | 52.4 | 67.5 |
| GaPro + SPFormer  | 53.5 | 72.2 |
| Sketchy-3DIS      | 65.8 | **83.1** |
| Ours + MAFT       | **65.9** | 74.9 |

Considering that AP25 primarily reflects coarse localization, Sketchy-3DIS’s strong AP25 performance is consistent with its design focus on robustness to noisy boxes. In contrast, Ours + MAFT achieves the best AP50, indicating that BEEP3D continues to produce precise and reliable pseudo-labels even under noisy box annotations.

---

> ### Author Response · Authors · 2025-12-04
> **Global response #2**
>
> **Q3.** Failure case analysis (Reviewer UCvp, v8Bp, WQrj)
>
> **A3.** We appreciate the reviewers’ suggestions to provide a more detailed analysis of failure cases. As shown in Appendix A3 of the revised manuscript, BEEP3D tends to fail in highly cluttered scenes where the bounding box of one instance fully overlaps with that of another. In such cases, the coarse box-level supervision provides insufficient spatial cues to distinguish the two instances, and BEEP3D inherits this inherent limitation. We believe this analysis clarifies the model’s behavior under challenging annotation conditions and highlights meaningful directions for future work.
>
>
> We sincerely hope these revisions effectively address the reviewers' concerns and contribute to improving the overall quality of our paper. Thank you once again for your thoughtful review and valuable feedback.
>
> Best regards,
>
> The Authors
>
> [1] Schult, Jonas, et al. "Mask3D: Mask Transformer for 3D Semantic Instance Segmentation." 2023 IEEE International Conference on Robotics and Automation (ICRA). IEEE, 2023.
>
> [2] Lai, Xin, et al. "Mask-attention-free transformer for 3d instance segmentation." Proceedings of the IEEE/CVF International Conference on Computer Vision. 2023.
>
> [3] Lu, Jiahao, et al. "Relation3D: enhancing relation modeling for point cloud instance segmentation." 2025 IEEE/CVF Conference on Computer Vision and Pattern Recognition (CVPR). IEEE, 2025.
>
> [4] Wang, Duanchu, et al. "Enhancing 3D Instance Segmentation With Dense Connection Decoder and Layer-Aware Fusion." IEEE Robotics and Automation Letters (2025).
>
> [5] Lu, Jiahao, et al. "Query refinement transformer for 3d instance segmentation." Proceedings of the IEEE/CVF International Conference on Computer Vision. 2023.
>
> [6] Kolodiazhnyi, Maxim, et al. "Oneformer3d: One transformer for unified point cloud segmentation." Proceedings of the IEEE/CVF Conference on Computer Vision and Pattern Recognition. 2024.
>
> [7] Deng, Qian, et al. "Sketchy Bounding-box Supervision for 3D Instance Segmentation." Proceedings of the Computer Vision and Pattern Recognition Conference. 2025.

---

### Meta-Review · Area_Chair_CukW · 2026-01-07

**Summary:**

The paper initially received mixed reviews, with scores of 4, 8, 4, and 4.

The main concerns raised by several reviewers include limitations of the transformer-based architecture, robustness to noisy box annotations, and failure case analysis. Other concerns, such as relatively limited AP improvements, a lack of broader evaluation across more transformer variants, and insufficient statistical significance, were also mentioned by different reviewers.

The authors' rebuttal provided further explanations and additional experimental results to address the issues raised. However, some concerns remain unresolved. In particular, the robustness to noisy box annotations (scaling and center perturbations) was not fully supported by the experimental results. Furthermore, the experiments with OneFormer3D did not adequately address the limited generality issue.

Given that the final reviews have not reached consensus and some primary concerns remain, the area chair recommends rejecting this paper.

**Reviewer Concerns:**

The issues of failure case analysis and insufficient statistical significance were addressed by the rebuttal.

However, the concern about robustness to noisy box annotations remains unresolved, and the experimental results with OneFormer3D do not clarify the issue of limited generality.

 - **Robustness to Noisy Box Annotations**

    The results from the authors' responses indicate that noisy bounding-box annotations are a significant concern compared with previous methods such as Sketchy-3DIS.

    - *Evaluation on noisy bounding-box annotations (S1, scaling)*
    | Method | AP50 | AP25 |
    | -------- | -------- | -------- |
    | Sketchy-3DIS | 65.8 | **83.1** |
    | Ours + MAFT | **65.9** | 74.9  |

    - *Evaluation on noisy bounding-box annotations (S2, center perturbation)*
    | Method | AP50 | AP25 |
    | -------- | -------- | -------- |
    | Sketchy-3DIS | **63.7** | **82.1** |
    | Ours + MAFT | 62.5 | 72.5 |


- **Limited Generality**

    The performance drop between "Fully" and "Box" with OneFormer3D is more significant than that with SPFormer and MAFT, suggesting a potential architecture-dependency issue.

    | Sup. | Method | AP | AP50 | AP25 |
    | -------- | -------- | -------- |  -------- |  -------- |
    | Fully | OneFormer3D | 59.3 | 78.1 | 86.4 |
    | Box | OneFormer3D + Ours | 55.8 | 74.7 | 84.6 |


- **Metric Regression**

    Regarding the "metric regression" issue, **Reviewer UCvp** noted
    > Furthermore, I cannot agree that the regression of (from 75.9 to 75.3) can be ignored (the authors argue that coarse metrics are "saturated"). A robust SOTA method should ideally improve (or at least maintain) detection recall.

    The area chair agrees with **Reviewer UCvp**, as the authors' argument is not sufficiently convincing.

**Reviewer Scores:**

- The follow-up responses from **Reviewer UCvp** after the rebuttal pointed out several remaining concerns and mentioned
    > I will maintain my original score and think that this paper did not meet the bar of ICLR.

    The authors further responded to those comments, but issues like limited generality and reliance on ideal priors remained unresolved.

    The area chair expects the reviewer to maintain a score of **4**.

- The area chair expects **Reviewer v8Bp** to maintain the score at **8**.

- The area chair expects **Reviewer bt6s** to maintain a score of **4**, as similar concerns from **Reviewer UCvp** are not adequately addressed.

- **Reviewer WQrj** provided feedback after the rebuttal and noted
    > If you can address these two points in a more concrete way, I am happy to reconsider my score in a positive direction.

    While the issue of "benefit of the EMA teacher" is likely addressed by the authors' response, the concern about "the ISBNet variants" remains unclear with the authors' brief explanation.

    The area chair expects the reviewer to either raise the score to **6** or maintain it at **4**.

---

### Decision · Program_Chairs · 2026-01-26

Reject